# MAnchors: Memorization-Based Acceleration of Anchors via Rule Reuse and Transformation

**Haonan Yu** [1 2]   **Junhao Liu** [1 2]   **Xin Zhang** [1 2]

## Abstract

Anchors is a popular local model-agnostic explanation technique whose applicability is limited by its computational inefficiency. To address this limitation, we propose a memorization-based framework that accelerates Anchors while preserving explanation fidelity and understandability. Our approach leverages the iterative nature of Anchors, which gradually refines explanations until they are precise enough for a given input, by storing and reusing intermediate results from prior explanations. Specifically, we maintain a memory of low-precision, high-coverage rules and introduce a rule transformation framework to adapt them to new inputs: the horizontal transformation adapts a retrieved explanation to the current input by replacing features, and the vertical transformation refines the general explanation until it is precise enough for the input. We evaluate our method across tabular, text, and image datasets, demonstrating that it significantly reduces explanation generation time while maintaining fidelity and understandability, thereby enabling the practical adoption of Anchors in time-sensitive applications.

## 1. Introduction

Anchors (Ribeiro et al., 2018) is a popular local model-agnostic explanation technique that generates rules forming sufficient conditions for a model prediction on a given input. It is local because it explains model behavior around a particular input, making it applicable to complex models in domains such as healthcare and finance, where understanding individual predictions is vital for decision-making and validation. It is model-agnostic because it does not rely on model internals.

These properties have led to the adoption of Anchors in many applications (Lopardo et al., 2022) (Jayakumar & Skandhakumar, 2022), such as explaining a triage-prediction system for COVID-19 to enable knowledge discovery about patient risk factors (Khanna et al., 2023), and analyzing Existing Vegetation Type maps to uncover insights for ecological patterns and land management (Ganji & Lin, 2023). However, Anchors suffers from a major limitation: high computational cost. Generating an explanation for a single input can take several hours in certain settings, severely hindering its deployment in time-sensitive or interactive applications. This issue becomes even more pronounced when explaining large language models (LLMs), where Anchors' iterative sampling procedure incurs not only substantial latency but also significant monetary cost due to repeated API calls.

In this paper, we aim to substantially improve the computational efficiency of Anchors without sacrificing explanation quality. Our approach is motivated by two key observations: 1) The majority of Anchors' runtime is spent on sampling perturbed variants of the input instance. 2) Anchors constructs explanations in an iterative fashion, gradually refining rules from general, high-coverage candidates to specific, high-precision anchors. Importantly, these intermediate rules already capture meaningful local information about the model's behavior.

Based on these observations, we propose a memorization-based acceleration strategy for Anchors. The core idea is to maintain a memory set that stores previously generated explanation rules together with their associated inputs. When a new instance requires explanation, the algorithm retrieves relevant rules from memory and reuses them as initialization points, thereby avoiding costly sampling from scratch.

However, a naive caching strategy faces two significant challenges. First, exact matches are rare in high-dimensional spaces, especially for text and image data, leading to low cache-hit rates. Second, storing complete explanations for every input would be costly. To mitigate these issues, we exploit the iterative structure of Anchors by storing only low-precision, high-coverage rules. These rules are more

---

[1]School of Computer Science, Peking University, Beijing, China [2]Key Lab of High Confidence Software Technologies (Peking University), Ministry of Education, Beijing, China. Correspondence to: Xin Zhang <xin@pku.edu.cn>.

general, allowing them to be retrieved for broader regions of the input space, even when exact matches do not exist.

This design introduces additional challenges. Because Anchors is inherently local, even high-coverage rules may not be directly applicable to neighboring inputs. As illustrated in Table 1, a cached explanation derived from sample $x_1$ may involve a different feature than those present in a new input $x_2$, despite the two explanations being semantically similar. Moreover, low-precision rules retrieved from memory may not satisfy the fidelity requirements expected from a final anchor explanation.

To address these challenges, we propose a rule transformation framework that adapts cached rules to new inputs. We define two forms of rule transformation: one is horizontal, transforming a rule for one input into a rule for another input by substituting the features; the other is vertical, transforming a general rule (rule with high coverage and low precision) into a specific rule (rule with low coverage and high precision) by strengthening it. Together, these transformations enable effective reuse while preserving the local fidelity guarantees of Anchors.

Our experiments demonstrate that our memorization-based approach achieves significant acceleration. Specifically, when applied to the explanation of the Llama 3 (8B), our approach yields an 8.74× speedup and reduces the required samples by 87%. Furthermore, by utilizing vertical transformation to refine cached results, the method maintains a level of fidelity consistent with the original Anchors algorithm.

## 2. Background

This section provides the necessary background knowledge that is integral to understanding our approach. We first describe the form of Anchors explanations and then its underlying algorithm.

### 2.1. Representation of Anchors' Explanations

We first introduce the notation used throughout the paper. Let $\mathcal{X}$ denote the input space and $\mathcal{Y}$ the label space. A target model is a function $f : \mathcal{X} \to \mathcal{Y}$. An input instance $x \in \mathcal{X}$ is represented as a tuple $x = (x_1, x_2, ..., x_n)$, where each $x_i$ denotes the $i$-th feature and $n$ is the total number of features. A predicate is a function that maps an input $x$ into a binary value, i.e., $p : \mathcal{X} \to \{0, 1\}$. A predicate constrains one feature to a specific value, denoted as $p_{i,v}(x) := \mathbf{1}[x_i = v]$, where $\mathbf{1}[\cdot]$ is the indicator function. We now formalize the objective optimized by Anchors.

**Definition 2.1** (Rule). For an input $x$ of $n$ features, let $\mathcal{R}_x$ denote the set of rules satisfied by $x$. A rule $r \in \mathcal{R}_x$ is a conjunction of predicates: $r := p_{a_1,v_1} \wedge p_{a_2,v_2} \wedge \cdots \wedge p_{a_m,v_m}, m \leq n$.

Similar to a predicate, we treat a rule as a function, i.e., $r(x) = 1$ when $\forall i \in [1, m].p_{a_i,v_i}(x) = 1$. We use $\mathcal{R}$ to denote a generic rule set.

**Definition 2.2** (Perturbation Distribution). Given $x \in \mathcal{X}$, let $D_x(\cdot)$ denote a probability distribution over perturbed instances $z \in \mathcal{X}$ near $x$. For any rule $r$ with positive coverage, the conditional distribution is

$$D_x(z \mid r) := \frac{D_x(z) \cdot \mathbf{1}[r(z) = 1]}{\mathbb{P}_{z \sim D_x}[r(z) = 1]}.$$

Next, we define the two most important performance metrics of Anchors as follows:

**Definition 2.3** (Precision). Given a rule $r$, the precision is:

$$\text{Precision}(r) := \mathbb{E}_{z \sim D_x(\cdot|r)} \left[\mathbf{1}[f(z) = f(x)]\right].$$

**Definition 2.4** (Coverage). The coverage of rule $r$ is:

$$\text{Coverage}(r) := \mathbb{E}_{z \sim D_x(\cdot)} \left[\mathbf{1}[r(z) = 1]\right].$$

Precision measures how reliably the rule preserves the target prediction among inputs that satisfy it, while coverage measures how large that input region is. Together, they measure how faithful the explanation is. Next, we formally define the output of Anchors.

**Definition 2.5** (Anchors Output). Let $\tau \in [0, 1]$ and $\delta \in [0, 1]$ be thresholds. The Anchors explanation rule for model $f$ and input $x$ is a rule covering $x$, meeting the precision threshold $\tau$ with probability at least $1 - \delta$ while maximizing coverage:

$$r_x \in \underset{r \in \mathcal{R}_p}{\arg\max} \text{Coverage}(r)$$

where $\mathcal{R}_p = \{\hat{r} \in \mathcal{R}_x \mid \mathbb{P}(\text{Precision}(\hat{r}) \geq \tau) \geq 1 - \delta\}$.

### 2.2. The Algorithm of Anchors

We next summarize the Anchors algorithm, since MAnchors modifies its search procedure.

As shown in Figure 1, the Anchors algorithm mainly consists of the following three steps:

- It generates a set of available predicates $\mathbb{P}$ based on the input $x$.

- It adds each predicate $p_{i,v}$ from the predicates set $\mathbb{P}$ that has not yet appeared in a rule named $r$ (which initially contains no predicates) to form the candidate rule set $\mathbb{R}_{next} = \{r \wedge p | p \in \mathbb{P} - r\}$. It uses the KL-LUCB algorithm (Kaufmann & Kalyanakrishnan, 2013) to determine the sample number $M$, and feeds $r$, $M$, and $x$ into the perturbation model to generate the neighborhood $D_x$ of the input $x$. Then it calculates *coverage* and *precision* of all the rules in rule set $\mathbb{R}_{next}$ using $D_x$.

| Input sample $x$ | Generated explanation $R_x$ | $f(x)$ |
|---|---|---|
| $x_1$ = This is the best movie that I've ever seen. | ("best") | positive |
| $x_2$ = He was really nice today and helped me a lot. | ("nice") | positive |

*Table 1.* Two semantically similar positive inputs can require different feature predicates.

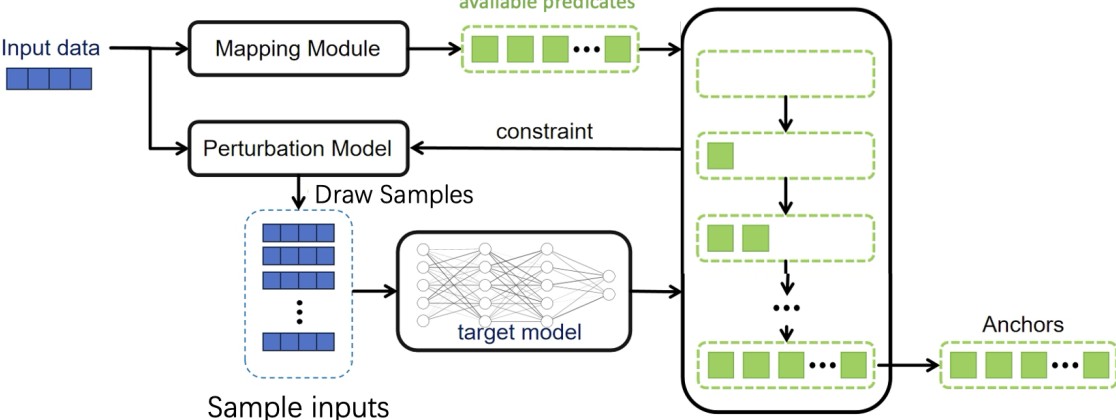

*Figure 1.* The overall workflow of Anchors.

- If the precision of any rule is no less than $\tau$, it returns a rule that has the maximum *coverage* among all rules satisfying the precision requirement. Otherwise, it sets $r$ to be the rule with the highest precision and continues to Step 2.

It can be seen that after each iteration, one predicate is added to the rule $r$. As the number of predicates in $r$ increases, the number of inputs it can cover will decrease, while the probability that the prediction results of the covered inputs are the same as the original input's will increase. In other words, during the iteration, the *coverage* will gradually decrease while the *precision* will gradually increase, and the rule $r$ will gradually transform from a "general" explanation to a "specific" explanation for a particular input.

## 3. Our Approach

To reduce Anchors' computational cost, our method adopts a memorization-based strategy designed to reuse previous computations and significantly reduce redundant sampling. The core of our approach involves maintaining a memory set (initially empty) that stores previously processed input samples alongside their corresponding intermediate results. When generating an explanation for a new instance, the system first performs a similarity search within this memory set. The workflow branches according to whether retrieval yields a memory hit:

- **Memory Miss**: If no similar sample is found, the system defaults to the standard Anchors procedure to generate an explanation from scratch. The resulting intermediate rule is then cached in the memory set for future use.

- **Memory Hit**: If a sufficiently similar sample exists, we bypass the full sampling process by applying two distinct transformations to the retrieved intermediate results:

  - A **horizontal transformation**, which modifies the intermediate rule to match the feature space of the new input.
  - A **vertical transformation**, which incrementally refines this adapted rule to meet a required precision threshold.

These transformations reduce the number of samples required after a memory hit and thereby accelerate explanation generation. Furthermore, even during a memory miss, the computational overhead of our retrieval mechanism remains several orders of magnitude lower than the sampling process of the original Anchors algorithm. We provide a detailed complexity analysis and discussion of this overhead in Section 3.5.

### 3.1. Overview

We use the image-classification example in Figure 2 to show the workflow of our method and how horizontal and vertical transformations operate. Anchors explain images as sets of superpixels that serve as sufficient conditions: if all are present, the model likely repeats its original prediction.

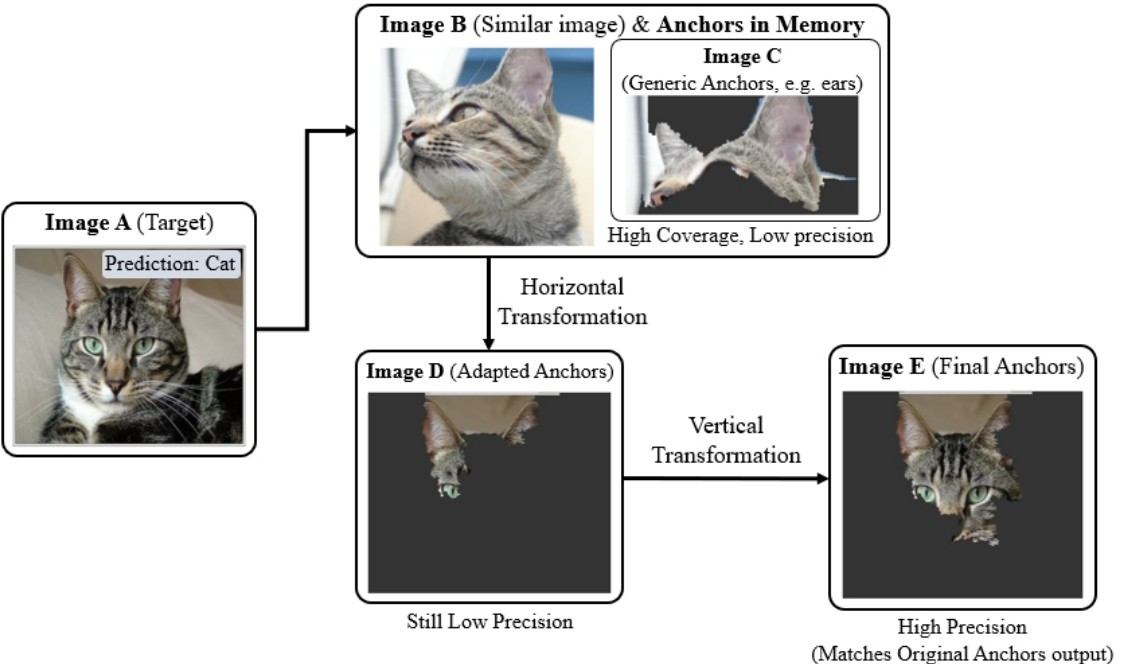

*Figure 2.* An example of our workflow for image.

Suppose Image A is classified as a cat. Our approach retrieves a similar image, Image B, with an intermediate anchor (Image C), often representing a generic feature like ears. To ensure generality, this intermediate anchor uses few superpixels, giving high coverage but low precision.

Because Image B's anchor differs in pixel-level representation, it cannot be applied directly to Image A. We first apply a *horizontal transformation* to adapt it, replacing its superpixels with visually similar ones in Image A to produce a new anchor (Image D) that covers relevant parts but still lacks precision. Next, a *vertical transformation* refines the explanation by adding superpixels until the desired precision is reached, yielding a final anchor (Image E) that matches the original Anchors output with fewer samples and faster computation. The following subsections describe the specific steps of our method.

### 3.2. Main Workflow

Algorithm 1 takes an input $x$ and model $f$, and returns an explanation rule. First, the most similar input $x_{best}$ in $\mathbb{M}$ is identified via multi-dimensional embeddings (Line 1), using the embedding method of Anchors' internal perturbation model. This ensures similar samples produce overlapping perturbations. The similarity is then quantified based on the distance between the embedded multi-dimensional vectors (Line 2) and evaluated against the threshold (Line 3). Finally, depending on the outcome of this comparison, the process branches into two cases: a memory miss (Line 4) or a memory hit (Line 7). In both cases, the resulting rule $r$ is

---

**Algorithm 1** MAnchors

**Input**: The input $\boldsymbol{x} = (\boldsymbol{x}_1, \boldsymbol{x}_2, ..., \boldsymbol{x}_n)$, the model to explain $f$

**Parameter**: The precision threshold $\tau_p$ for output, the precision threshold $\tau_{p_{mid}}$ for intermediate rules and the similarity threshold $\tau_{sim}$

**Output**: The explanation $r$

1: $(\boldsymbol{x}_{best}, r_{mid}) := FindMostSimilar(x, \mathbb{M})$
2: $similarity := CalculateSimilarity(\boldsymbol{x}, \boldsymbol{x}_{best})$
3: **if** $similarity < \tau_{sim}$ **then**
4:    $\# Case: Memory\ Miss$
5:    $r := ProcessMemoryMiss(\boldsymbol{x}, f, \tau_p, \tau_{p_{mid}})$
6: **else**
7:    $\# Case: Memory\ Hit$
8:    $r := ProcessMemoryHit(\boldsymbol{x}, r_{mid}, f, \tau_p)$
9: **end if**
10: return $r$

---

returned as the explanation (Line 10).

### 3.3. Memory Miss

Algorithm 2 outlines the process when a memory miss occurs, which takes $\boldsymbol{x}$ and $f$, and returns an explanation rule. The parameter $\tau_p$ specifies the precision requirement for output. The parameter $\tau_{p_{mid}}$ specifies the precision requirement for the intermediate rule, which is lower than $\tau_p$ to get a low-precision but more-coverage rule.

**Algorithm 2** ProcessMemoryMiss

---

**Input**: The input $\boldsymbol{x} = (\boldsymbol{x}_1, \boldsymbol{x}_2, ..., \boldsymbol{x}_n)$, the model to explain $f$

**Parameter**: The precision threshold $\tau_p$ for output, the precision threshold $\tau_{p_{mid}}$ for intermediate rules and the probability threshold $\delta$

**Output**: The explanation $r$

1: $r_{mid}, r := Anchors(f, \boldsymbol{x}, \tau_p, \tau_{p_{mid}}, \delta)$
2: $\mathbb{M} := \mathbb{M} \cup \{(\boldsymbol{x}, r_{mid})\}$
3: return $r$

---

**Algorithm 3** ProcessMemoryHit

---

**Input**: The input $\boldsymbol{x} = (\boldsymbol{x}_1, \boldsymbol{x}_2, ..., \boldsymbol{x}_n)$, the intermediate rule $r_{mid}$ and the model to explain $f$

**Parameter**: The precision threshold $\tau_p$ and the probability threshold $\delta$

**Output**: The explanation $r$

1: $\# Horizontal\ Transformation$
2: $r_1 := \emptyset$
3: **for** $p_{i,v}$ **in** $r_{mid}$ **do**
4:     $j := ChooseAny(arg\ min_{j \in \{1,2,..,n\}}(Dist(v, \boldsymbol{x}_j)))$
5:     $r_1 := r_1 \wedge \{p_{j,\boldsymbol{x}_j}\}$
6: **end for**
7: $\# Vertical\ Transformation$
8: $r = r_1$
9: **repeat**
10:     $\mathbb{R}_{next} := \emptyset$
11:     **for** $i = 1$ **to** n **do**
12:        **if** $p_{i,\boldsymbol{x}_i} \notin r$ **then**
13:           $\mathbb{R}_{next} := \mathbb{R}_{next} \cup \{r \wedge \{p_{i,\boldsymbol{x}_i}\}\}$
14:        **end if**
15:     **end for**
16:     $r := ChooseAny(arg\ max_{r_i \in \mathbb{R}_{next}}(precision_{\boldsymbol{x},f}(r_i)))$
17: **until** $P(precision_{\boldsymbol{x},f}(r) \geq \tau) \geq 1 - \delta$
18: return $ChooseAny(arg\ max_{r_i \in \mathbb{R}_{next}}(coverage_{\boldsymbol{x},f}(r_i)))$

---

Initially, the Anchors algorithm is invoked to generate rules $r$ and $r_{mid}$, which serve as explanations for the model $f$ on instance $x$ with precision levels of at least $\tau_p$ and $\tau_{p_{mid}}$(Line 1), respectively. $r_{mid}$ represents an intermediate result produced during the iterative process of generating $r$. This intermediate rule, paired with its corresponding input $\boldsymbol{x}$, is then integrated into the memory set $\mathbb{M}$ (Line 2). Finally, the rule $r$ is returned as an explanation (Line 3).

### 3.4. Memory Hit

Algorithm 3 outlines the process when a memory hit occurs. It takes an input $\boldsymbol{x}$, a model $f$, the intermediate rule $r_{mid}$ as input. Additionally, it includes a parameter $\tau_p$ that controls the precision of the Anchors explanation. The output of the Algorithm 3 is an explanation $r$ for the input $\boldsymbol{x}$ and the

model $f$. Next, we discuss the details of Algorithm 3 and provide formal definitions for HT and VT.

**Horizontal transformation.** From Line 1 to Line 6 is the horizontal transformation (HT). The HT maps the predicates from the intermediate rule $r_{mid}$ onto the predicates formed by the features in the input $\boldsymbol{x}$ that are most similar to them. We first give the mathematical definition of HT, and then introduce its implementation:

**Definition 3.1** (Horizontal Transformation). Let $r_1 = p_{a_1,\boldsymbol{x}_{a_1}} \wedge p_{a_2,\boldsymbol{x}_{a_2}} \wedge \cdots \wedge p_{a_n,\boldsymbol{x}_{a_n}}$ be a rule derived from a memorized input $\boldsymbol{x}$. The function HT $: \mathbb{R}_{\boldsymbol{x}} \to \mathbb{R}_{\boldsymbol{x}'}$ maps $r_1$ to a new rule $r_2 = p_{b_1,\boldsymbol{x}'_{b_1}} \wedge p_{b_2,\boldsymbol{x}'_{b_2}} \wedge \cdots \wedge p_{b_m,\boldsymbol{x}'_{b_m}}$ for current input $\boldsymbol{x}'$, where $m \leq n$ and each predicate $p_{i,\boldsymbol{x}_i}$ in $r_1$ is transformed into a predicate $p_{j,\boldsymbol{x}'_j}$ in $r_2$ such that $\boldsymbol{x}'_j$ is the most similar feature to $\boldsymbol{x}_i$ based on the following criterion:

$$\text{Dist}(\boldsymbol{x}_i, \boldsymbol{x}'_j) \leq \text{Dist}(\boldsymbol{x}_i, \boldsymbol{x}'_k), \quad \forall k \in \{1, \ldots, m\}$$

The function $\text{Dist}(\cdot, \cdot)$ represents a distance metric defined in the perturbation space. For tabular data, it represents the absolute value of the difference between two numbers. For text data, it represents the distance between two words in the semantic space generated by a fine-tuned BERT (Devlin et al., 2019). For image data, it represents the distance between the vectors of two superpixels after embedding by the explaining models. The design of the Dist is intended to reflect the similarity between two features within the perturbation space; that is, the smaller the Dist between two features, the more likely they are to be substituted for one another during perturbation.

Then we introduce the detailed steps of HT: In Line 3, the algorithm enumerates each predicate $p_{i,v}$ in the intermediate rule $r_{mid}$. In Line 4, the algorithm identifies the most similar feature in the input $\boldsymbol{x}_j$ based on the value $v$ in $p_{i,v}$. At last, in Line 5, the algorithm adds the predicate $p_{i,similar\_feature}$ to the rule $r_1$.

**Vertical transformation.** From Line 7 to Line 17 is the vertical transformation (VT). This process refines a general rule—which typically has high coverage but low precision—into a more specific rule with low coverage and high precision. The formal definition of VT is as follows:

**Definition 3.2** (Vertical Transformation). Let $\tau \in [0, 1]$, $\delta \in [0, 1]$ be thresholds. The function $VT : \mathbb{R}_{\boldsymbol{x}} \to \mathbb{R}_{\boldsymbol{x}}$ maps a basic rule $r_1$ into a rule covering $\boldsymbol{x}$, implying $r_1$, meeting the precision threshold $\tau$ with high probability $\delta$ while maximizing the coverage:

$$r_{\boldsymbol{x}} \in \text{argmax}_{r \in R_p} Coverage(r)$$

where $R_p = \{\hat{r} \mid r(\boldsymbol{x}) = 1 \wedge P(Precision(\hat{r}) \geq \tau) \geq 1 - \delta \wedge \hat{r} \in \mathbb{R} \wedge \hat{r} \Rightarrow r_1\}$

This process works by continuously adding new predicates to the current rule $r$. At the start of each iteration (Line 10), the candidate rule set is initialized as empty. In Line 11, the algorithm enumerates all features of input $\boldsymbol{x}_i$. Lines 12-13 add a new predicate $p_{i,\boldsymbol{x}_i}$, not already in $r_1$, to form candidate rules. In Line 16, perturbation sampling selects the candidate with the highest precision as the new rule $r$ for the next iteration. This repeats until $Precision(r) \geq \tau$ with high probability, after which the rule with the highest *coverage* is output (Line 17).

Thus, we align with Anchors' precision goals and ensure that explanations can be effectively adapted to specific contexts without extensive recalculations.

**Relation to Beam Search and KL-LUCB.** Vertical transformation preserves the beam-search structure of Anchors rather than replacing it: it starts from an adapted intermediate rule instead of the empty rule and continues the same search until the target precision is reached. Therefore, MAnchors remains compatible with beam search while reducing the number of search iterations. Since each iteration invokes KL-LUCB to estimate candidate precision, fewer iterations directly reduce total sampling.

**Cost of Horizontal Transformation.** If the retrieved rule contains $n_1$ predicates and the current input has $n_2$ candidate features, HT requires at most $n_1 n_2$ feature-similarity evaluations. We use the same perturbation model already employed by Anchors to embed features for the distance function, and then remove low-contribution predicates by light sampling after HT. Thus, imperfect similarity estimates are filtered before VT, while HT remains negligible relative to repeated model queries in Anchors.

### 3.5. Theoretical Overhead Analysis

We demonstrate that our approach introduces virtually no additional time cost compared to the original Anchors method. The analysis focuses on the time complexity of generating an explanation for a single input. For the original Anchors method, we assume the final explanation requires $k$ predicates, a single query to the target model has time complexity $O(T)$, the input $\boldsymbol{x}$ has $n$ features, the size of $\mathbb{M}$ is $m$, and the average complexity of one KL-LUCB run is $O(n \log(n) \log(n/\tau))$ (Kaufmann et al., 2016). Thus, the overall average complexity of Anchors is $O(k \cdot n \log(n) \log(n/\tau) \cdot T)$. Our method introduces a memory-based search using a k-d tree (Bentley, 1975). The worst-case complexity for a search and insertion is $O(m)$, where $m$ is the size of the memory set $\mathbb{M}$. Therefore, the total complexity of our method when a memory miss occurs is:

$$O(m + k \cdot n \log(n) \log(n/\tau) \cdot T)$$

In practice, the overhead $O(m)$ is asymptotically negligible for two primary reasons: 1)Since $T$ represents the cost of a forward pass through a target model (often a deep neural network with millions of parameters), we can safely assume $T \gg m$ in most practical AI applications; 2)Anchors requires thousands of such queries across its iterative search.

Even if a memory miss occurs, a k-d tree operation is measured in milliseconds in our setting, while the $k$ iterations of Anchors can take minutes or hours. Thus, the added lookup cost is negligible in practice.

**Space Overhead.** MAnchors stores (i) a k-d tree over cached embeddings, (ii) sample indices, and (iii) the feature subsets of reusable intermediate anchors. With memory size $m$, embedding dimension $d$, and average rule length $\ell$, the overhead is $O(md) + O(m) + O(m\ell)$. This is a shared cache of reusable rules rather than one final explanation per input. Empirically, excluding model and dataset memory, MAnchors uses 279.97 MB on average versus 276.81 MB for Anchors, a small increase relative to the observed speedups.

## 4. Experiment

In this section, we aim to empirically evaluate the effectiveness of our method compared to the original Anchors algorithm. Our experiments are designed to answer the following key questions:

- **Efficiency Improvement:** To what extent does our method improve the efficiency of Anchors? Specifically, how much time and sampling cost are reduced?

- **Fidelity Preservation:** Does our accelerated method preserve the fidelity and understandability of the original Anchors explanations, in terms of precision and coverage?

To comprehensively address these questions, we evaluate our approach on tabular, text, and image data with multiple models and datasets.

### 4.1. Experiment Setup

For tabular data, we selected income prediction as the target task; for text data, we selected sentiment analysis as the target task; for image data, we selected image classification as the target task. Datasets and models follow the original Anchors setup when possible.

Across all tasks, we set $\tau_p = 0.95$ and $\delta = 0.6$ to remain consistent with the default parameters of the Anchors method. The additional parameters introduced by our approach are set to $\tau_{p_{mid}} = 0.8$ and $\tau_{sim} = 0.6$. These values

were identified via grid search as the optimal configuration for maximizing speedup while ensuring that the fidelity and understandability of the generated explanations remain within an acceptable range (with a mean variation of less than 10% in coverage and 25% in length, respectively). The order of the test dataset was randomly shuffled, and the random seed is 42.

**Income Prediction.** The income prediction models take the numerical values of a person's multiple features (such as age, education level, race, etc.) as input and output whether their annual income will exceed \$50k or not, i.e. $f : X \rightarrow \{0, 1\}$, where $X := \prod_{i=1}^{k} F_i$ is the input domain, and $k$ represents the number of features, $F_i$ represents the value of i-th feature. We used the random forest (RF) (Breiman, 2001), gradient boosted trees (GBT) (Chen, 2016) and a 3-layers neural network (NN) (Rumelhart et al., 1986) as models to explain. We used these models to predict the data of 1,600 individuals from the Adult dataset (Becker & Kohavi, 1996), and explained the local behavior of the models around each input item in tabular.

**Sentiment Analysis.** Sentiment analysis models take a text sequence as input and predict whether its sentiment is positive or negative, i.e. $f : X \rightarrow \{0, 1\}$, where $X := \bigcup_{i=1}^{\infty} W^i$ is the input domain, and $W$ is the vocabulary set. We used the random forest (RF) (Breiman, 2001) and the 8B version of Llama3.0 (Llama) (AI@Meta, 2024) as the target models. We used these models to predict 2,100 comments from the RT-Polarity dataset (Pang & Lee, 2005), and explained the local behavior of the models around each input text.

**Image Classification.** Image classification models take an image as input and predict its category, i.e., $f : X \rightarrow \{0, 1, ..., m\}$, where $m$ is the number of categories and $X := \mathbb{R}^{3 \times h \times w}$ is the input domain, with $h$ and $w$ denoting image height and width. We used a pre-trained YOLOv8 (Glenn et al., 2023) to classify images from an ImageNet (Deng et al., 2009) subset. Because the full dataset is too large and Anchors is time-consuming, we randomly selected 10 categories and used 2000 images from these 10 categories as the test dataset.

All experiments were conducted on a server with 4 high-performance GPUs (10,000+ CUDA cores and 24GB memory each) and 256GB system memory. The full run took about 300 hours.

### 4.2. Efficiency Improvement

#### 4.2.1. EVALUATION METRICS

To quantify the performance of our accelerated Anchors method, we measured the following metrics:

**Speedup**: The speedup of our method compared to the original Anchors, calculated as (Time taken by the original Anchors/ Time taken by our method).

Let time$(\cdot)$ denote computation time, $f$ represent the model to explain, $\tau$ denote the precision threshold and $X$ denote the test dataset. Then, the **Speedup** can be expressed as:

$$SP = \frac{\sum_{x \in X} \text{time}(\text{Anchors}(f, x, \tau_p))}{\sum_{x \in X} \text{time}(\text{MAnchors}(f, x, \tau_p, \tau_{p_{mid}}))}$$

Where Anchors$(f, x, \tau_p)$ refers to the baseline process for interpreting $f$ applied to $x$ with precision threshold $\tau_p$, and MAnchors$(f, x, \tau_p, \tau_{p_{mid}})$ represents our accelerated method.

In other words, the higher the $SP$, the more evident the optimization effect of our method.

**Sampling Reduction Ratio**: This metric is calculated as $1 - \frac{\text{Sampling count of our method}}{\text{Sampling count of Anchors}} \times 100\%$. It provides a clearer measure of the reduction in API calls. A higher ratio indicates stronger optimization.

#### 4.2.2. EVALUATION RESULT

Table 2 shows the Speedup and sampling reduction ratios across three tasks with different models.

For sentiment analysis with Llama, our method achieved the highest acceleration across all models and tasks. The speedup was 8.74×, with an 87% sampling reduction. It is important to note that the absolute runtime for Llama remained the highest across all experiments. This matches our main objective: acceleration is most valuable when the original Anchors overhead is prohibitive.

For other tasks and models, the acceleration remained effective. For the Income Prediction task, our method achieved a speedup of 2.11×, 1.87× and 1.76× for RF, NN and GBT, respectively. For the sentiment analysis task and RF model, our method achieved a speedup of 1.69×. For the image classification task and YOLOv8 model, our method achieved a speedup of 1.92×. We attribute the variance in acceleration performance to two primary factors: model efficiency and dataset scale. Unlike Llama, other models exhibit higher computational efficiency where the sampling process—the specific target of our optimization—occupies a smaller fraction of the total execution time. Furthermore, for the limited size of certain test sets (e.g., YOLOv8 with only 200 images per category), cold start overhead accounts for a larger proportion of the total execution time, thereby diluting the overall measurable acceleration.

More detailed graphs showing the average runtime and samples of our method and Anchors as the number of inputs increases are provided in the Appendix. These results show that MAnchors completes the cold-start phase within 100 inputs per category for each type of task.

| Models to explain | Anchors Avg. Time (s) | MAnchors Avg. Time (s) | Speedup | Sampling Reduction Ratio |
|---|---|---|---|---|
| | | Income Prediction | | |
| RF | 8.51 ± 0.21 | 4.02 ± 0.16 | 2.11× | 54% |
| NN | 0.23 ± 0.01 | 0.12 ± 0.01 | 1.87× | 51% |
| GBT | 0.18 ± 0.01 | 0.10 ± 0.01 | 1.76× | 46% |
| | | Sentiment Analysis | | |
| RF | 32.10 ± 3.91 | 18.92 ± 3.14 | 1.69× | 32% |
| Llama | 513.06 ± 90.80 | 58.70 ± 21.70 | 8.74× | 87% |
| | | Image Classification | | |
| YOLOv8 | 61.34 ± 6.21 | 31.91 ± 5.06 | 1.92× | 47% |

*Table 2.* Acceleration effects of our approach on the three tasks (mean ± 95% confidence interval).

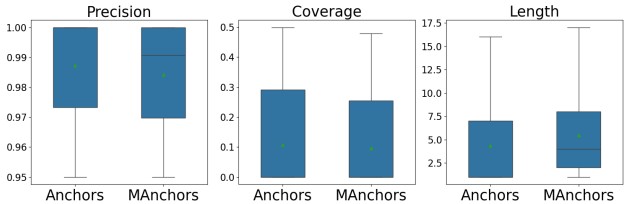

*Figure 3.* Fidelity and explanation length for Llama on sentiment analysis.

In summary, our method significantly accelerates Anchors, especially when model queries are expensive. It is also compatible with an offline pre-sampling deployment: setting $\tau_{sim} = 0$ recovers a pre-populated cache, while the default online strategy avoids the heavy upfront cost of explaining a large representative set before deployment.

### 4.3. Fidelity Preservation

We use **coverage** and **precision** from Section 2.1 to evaluate the fidelity of our method. Additionally, we measure predicate **length** as a standard proxy for understandability; this metric is limited, but rule-based Anchor explanations are widely used because their if–then form is directly readable by humans. Figure 3 shows fidelity and understandability results for Llama's explanations on the sentiment analysis task, which is the case where our acceleration effect is most significant. (Complete results are provided in the appendix.) Our method reaches the target precision by iteratively adding predicates until the threshold $\tau$ is met, yielding precision nearly identical to Anchors. The coverage decreased slightly from an average of 0.10 (±0.014) to 0.09 (±0.013), and the length increased from an average of 4.34 (±0.40) to 5.39 (±0.41). Overall, both fidelity and understandability remained relatively stable.

The slight degradation arises when retrieved inputs are not sufficiently similar to the current input, so HT contributes

predicates that are less aligned with the target neighborhood and VT must add extra predicates to recover precision. In practice, users can mitigate this trade-off by increasing $\tau_{sim}$, albeit with longer cold-start latency.

### 4.4. Parameter Sensitivity

The parameter study in Appendix C confirms the speed–quality trade-off motivating our default setting. Across the evaluated RF, NN, and GBT models, larger $\tau_{sim}$ typically improves coverage because retrieved rules are better aligned with the online input, while larger $\tau_{p_{mid}}$ tends to save more predicates and reduce runtime. We observe a stable operating region around $\tau_{sim} \in [0.6, 0.8]$ and $\tau_{p_{mid}} \in [0.7, 0.8]$, which balances acceleration with fidelity and explanation length.

### 4.5. Additional LLM Evaluation

To test whether the gains extend beyond one LLM configuration, we further evaluate Gemma-2-2B on SST-2 and AG News. MAnchors achieves 4.56× speedup on SST-2 (269.96 s to 59.17 s) with 78% sampling reduction, and 2.91× on AG News (259.31 s to 88.96 s) with 66% sampling reduction. Coverage changes remain modest (0.17 to 0.13 on SST-2; 0.50 to 0.48 on AG News), supporting generalization across LLM architectures and text tasks.

## 5. Related Work

Our work relates to model-agnostic explanation techniques and methods for improving their efficiency. These methods treat the target model as a black box. Prominent methods include Local Interpretable Model-agnostic Explanations (LIME) (Ribeiro et al., 2016), SHapley Additive exPlanations (SHAP) (Lundberg & Lee, 2017), Partial Dependence Plots (PDP) (Friedman, 2001), Individual Conditional Expectation (ICE) plots (Apley & Zhu, 2020), LORE (Guidotti

et al., 2018), and Anchors (Ribeiro et al., 2018). Among them, SHAP and Anchors achieve high fidelity and understandability but suffer from very low computational efficiency, which is a major drawback (Cummins et al., 2024).

Covert et al. (Covert et al., 2024) utilized stochastic amortization to speed up feature and data attribution via training a surrogate model to predict the explanation. However, their work focuses specifically on attribution-based methods (such as SHAP and LIME). Consequently, it does not address acceleration for rule-based methods such as Anchors. Furthermore, their method requires training a model to fit the explanation method, while our method does not involve an offline training phase.

Mor et al.(Mor et al., 2024) proposed a framework to accelerate global Anchor aggregations. While Anchors generate local explanations, global insights require executing the algorithm across entire datasets, which is computationally expensive. Mor et al. optimize this by tuning internal parameters and using candidate filtering. However, their acceleration method reduces the precision requirements, and the candidate filtering operation skips the explaining process for some inputs. Therefore, their method does not accelerate generation of a local explanation for a single input. Several recent studies extend Anchors in directions complementary to ours. ReX (Liu & Zhang, 2025) incorporates temporal information into model-agnostic local explanations, making Anchor-style reasoning applicable when decisions depend on evolving input histories rather than static instances. Chiu et al. (Chiu et al., 2023) combine Anchors with temporal logic and Monte Carlo Tree Search to explain dynamic decision systems, broadening the rule language from feature conjunctions to temporally structured conditions. ConLUX (Liu et al., 2024) lifts local explanations from raw features to human-interpretable concepts, seeking a unified representation that better matches user understanding. Lopardo et al. (Lopardo et al., 2023) analyze the behavior of Anchors for text data in depth, clarifying when its perturbation and rule-search mechanisms produce stable or unstable explanations. These works expand the expressiveness, domains, or understanding of Anchor-style explanations; MAnchors addresses an orthogonal bottleneck by accelerating the generation of local Anchor explanations while preserving the original rule form.

## 6. Limitations and Future Work

MAnchors trades a small amount of memory for lower latency and may slightly reduce coverage when retrieved rules are imperfectly aligned with a new input. This trade-off matters in high-stakes settings where exact fidelity is paramount, and post-hoc explanations for black-box models should complement rather than replace domain validation or interpretable modeling when the latter is feasible. Fu-

ture work will study learned alignment between memory and online inputs, larger vision-transformer evaluations, and richer human-centered understandability metrics beyond rule length.

## 7. Conclusion

In this paper, we address the significant computational bottleneck of the Anchors explanation technique, which has traditionally hindered its deployment in time-sensitive and large-scale applications. By introducing a memorization-based acceleration strategy, we avoid repeatedly sampling from scratch. Our dual-transformation framework reuses generalized, high-coverage rules across diverse input spaces through horizontal feature adaptation and vertical precision refinement. This approach mitigates the challenges of high-dimensional data and low cache hit rates, ensuring that the local fidelity inherent to the original Anchors algorithm is preserved while drastically reducing the iterative sampling overhead.

Our experimental results, particularly when applied to LLMs (Llama 3, 8B), demonstrate the practical efficacy of this method. These findings suggest that caching and transforming intermediate rules is a viable path toward making high-quality model-agnostic explanations feasible for modern large language models. By significantly lowering the latency and monetary costs associated with repeated sampling, this work paves the way for broader adoption of interpretable AI in interactive and resource-constrained environments.

## Acknowledgements

We thank the reviewers for their constructive feedback, which helped improve the paper. This work was sponsored by the National Natural Science Foundation of China (NSFC) under Grant No. W2411051.

## Impact Statement

This paper presents work whose goal is to advance the field of Machine Learning. There are many potential societal consequences of our work, none of which we feel must be specifically highlighted here.

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

## A. Experimental Compute Resources

All the experiments were conducted on a server with 4 high-performance GPUs, each providing up to 10,000+ CUDA cores and 24GB of dedicated GPU memory, combined with 256GB system memory. The complete experiments took approximately 300 hours to run.

## B. The Use of Large Language Models (LLMs)

In this paper, large language models were used solely for minor language polishing. No large language model was involved in developing the core ideas, methods, or writing of the main content.

## C. Parameter Sensitivity

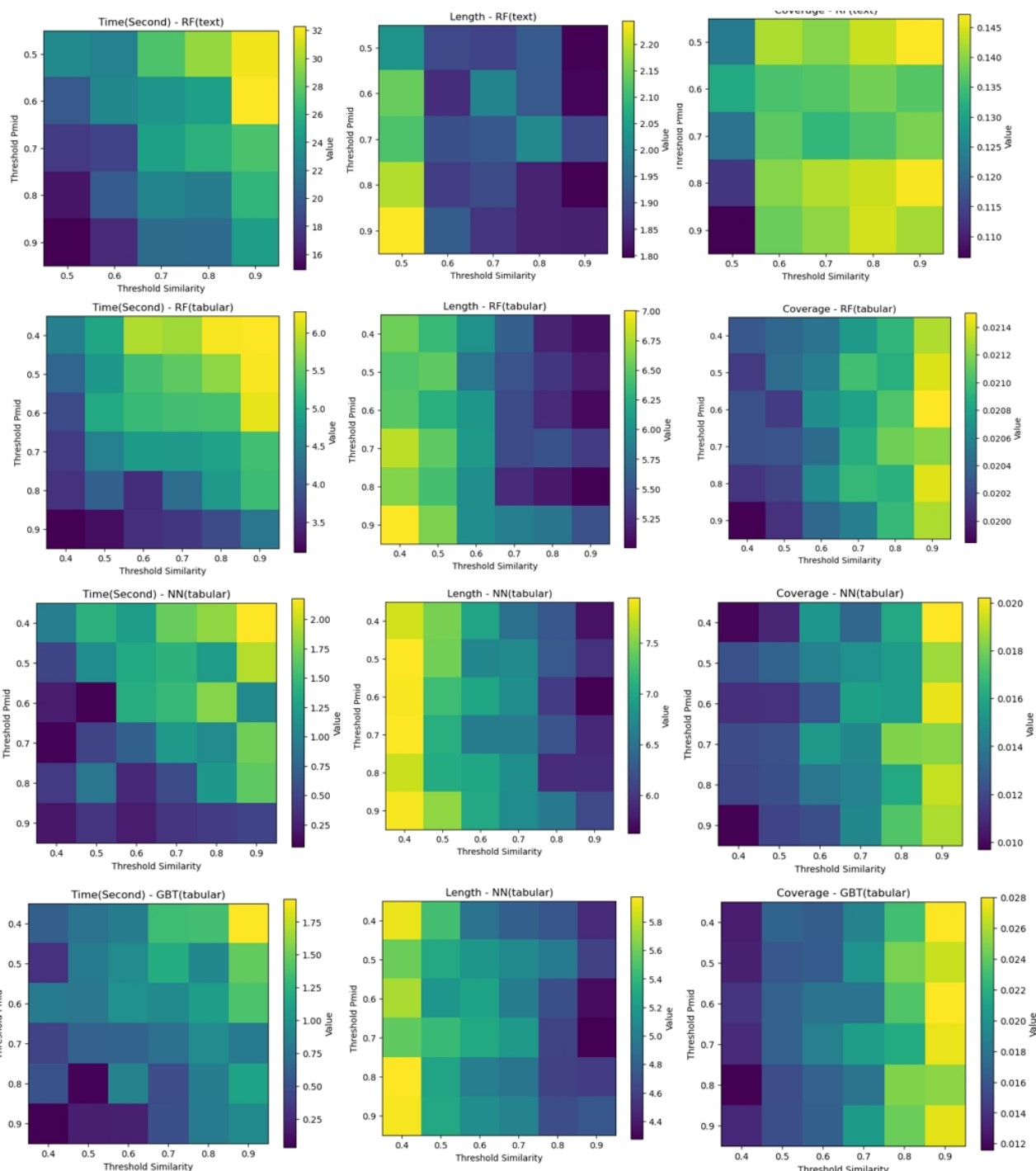

*Figure 4.* Sensitivity to $\tau_{sim}$ and $\tau_{p_{mid}}$ on representative models. Larger $\tau_{sim}$ generally improves coverage but increases runtime; larger $\tau_{p_{mid}}$ often shortens runtime with a mild coverage cost.

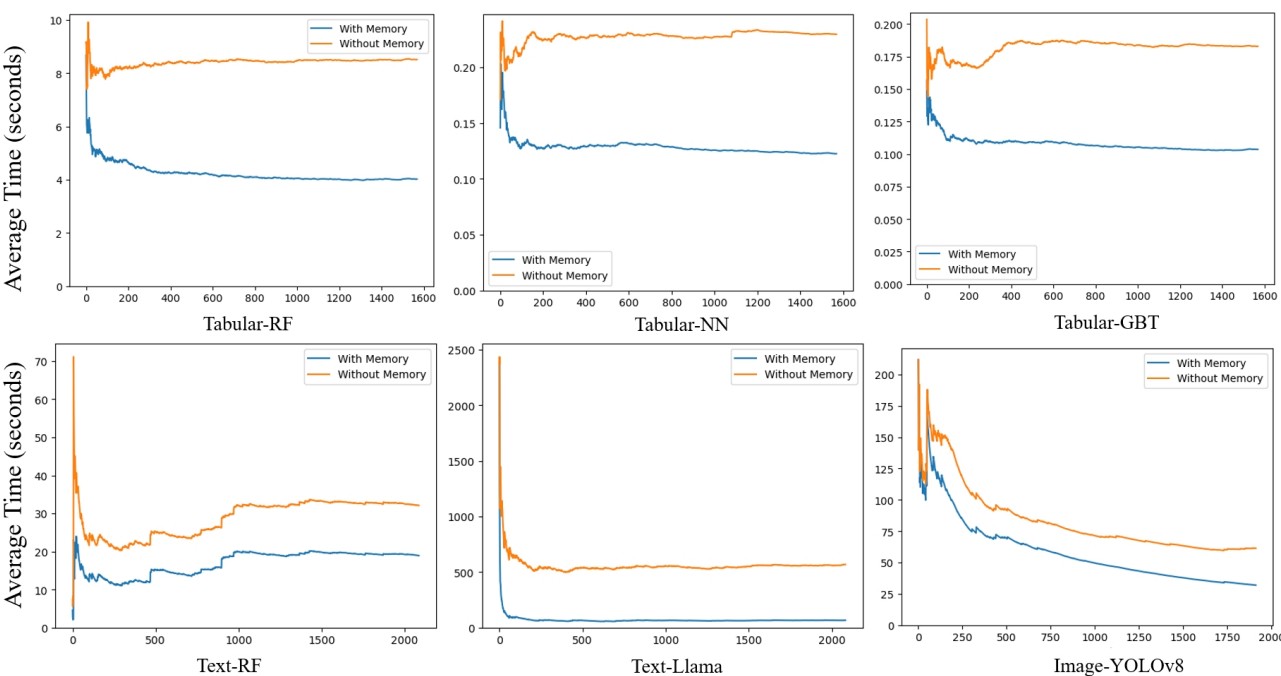

*Figure 5.* Average runtime of MAnchors and Anchors as the number of explained inputs increases.

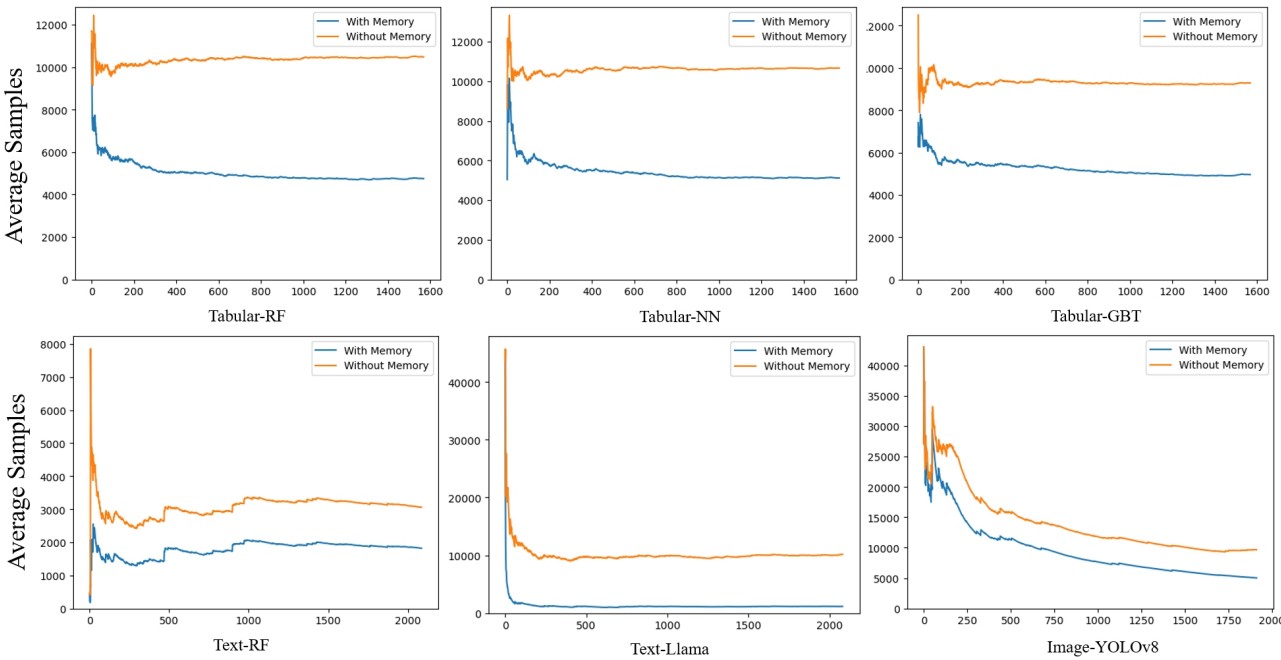

*Figure 6.* Average sample count of MAnchors and Anchors as the number of explained inputs increases.

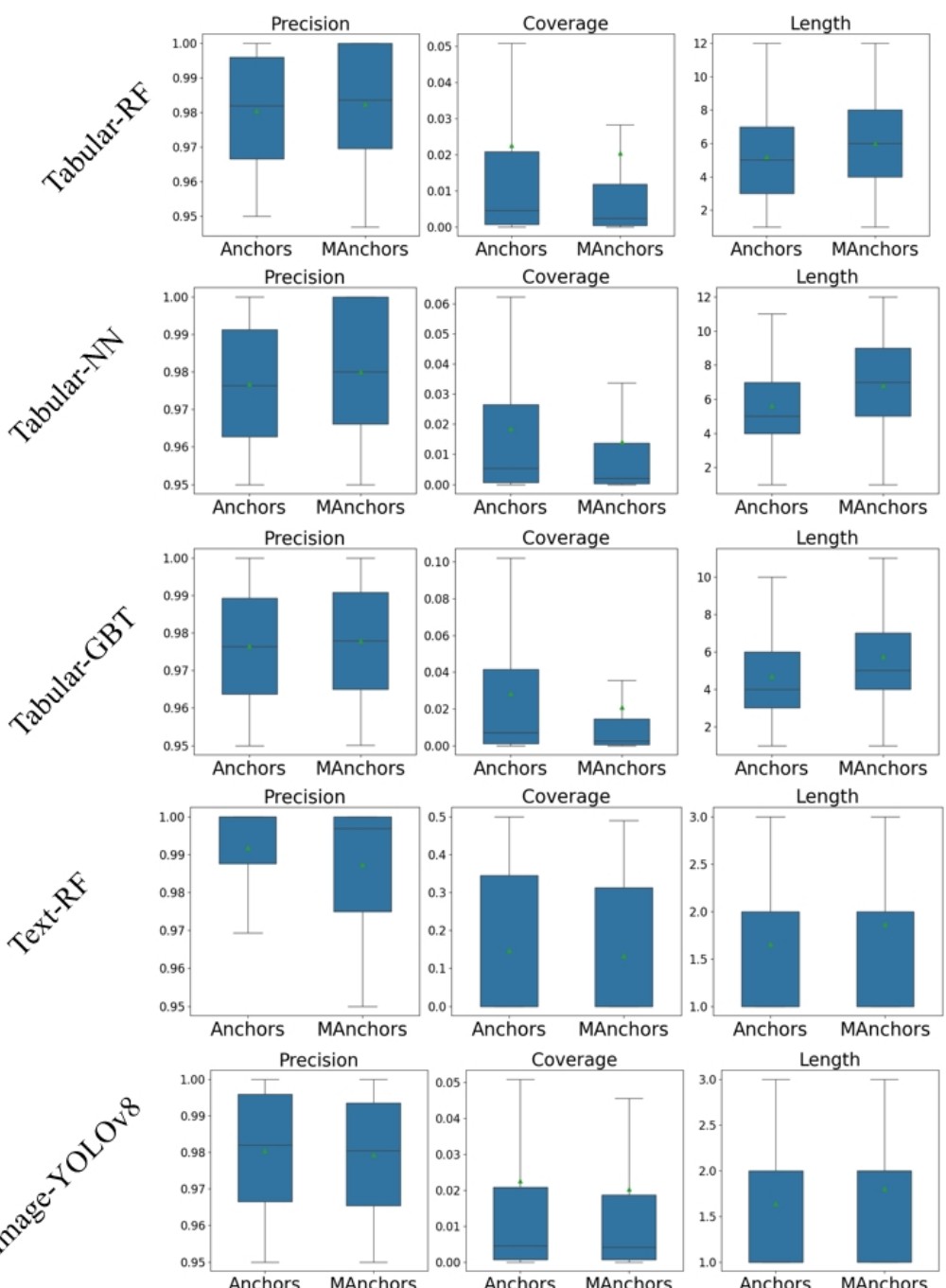

*Figure 7.* Fidelity and explanation-length results for MAnchors and Anchors across the remaining evaluated models.

