# OpenReview forum: "MAnchors: Memorization-Based Acceleration of Anchors via Rule Reuse and Transformation"
_ICML.cc/2026/Conference — ICML 2026 regular_

### Official Review · Reviewer_zjXL · 2026-02-28

**Soundness:** 3
**Presentation:** 3
**Significance:** 3
**Originality:** 3
**Overall Recommendation:** 5
**Confidence:** 4

**Summary:**

This paper presents a modified version of the Anchors algorithm by Ribeiro et al., with the main goal of improving its computational performance. The authors use caching of past computed rules to speed up rule generation for new instances. Caching is performed on high-coverage intermediate rules, not the final output, which can later be retrieved for similar inputs. The retrieved rules are modified to fit the new instance (horizontally) and to increase their precision (vertically). The authors perform experiments on tabular, text and image datasets and demonstrate significant speed-ups using their method at a small cost on other metrics (precision, coverage, length).

**Compliance With Llm Reviewing Policy:**

Affirmed.

**Final Justification:**

The rebuttal addressed my concerns and I have raised my score.

**Key Questions For Authors:**

1. Will you commit to providing a public implementation of your algorithm and experiments?

2. Can you confirm you will correct the unnormalized probability measure in Definition 2.2 and the input domain definition in Section 4.1? If you believe the current formulations are mathematically sound, please provide a justification.

3. Would you consider adding a small ablation study on the effects of τ_sim?

I am very willing to increase my score if these questions are adequately addressed.

**Limitations:**

I would like to see a small discussion on the impacts of explaining black-box models, instead of using interpretable models. There are many published works on how explanations can mislead, e.g., "Stop explaining black box machine learning models for high stakes decisions and use interpretable models instead" by Cynthia Rudin, and a small discussion on the potential negative impact of black-box explanation methods is very common in publications like this one.

**Strengths And Weaknesses:**

Strengths:

- I believe the method introduced by the authors will be certainly useful to any practitioners using the Anchors algorithm and could potentially widen its adoption as it addresses one of its major weaknessess, its computational inefficiency

- The experiments presented adequately support the authors' main claims

- The paper is well-written and easy to read


Weaknesses:

In my opinion the paper has 3 major weaknesses. Addressing these will greatly increase its quality and impact.

1. No code is provided with the submission. No reference is made in the text about providing a public implementation in the future. This is arguably the major drawback of this work, both in terms of supporting the authors' claims and in terms of future adoption.

2. The technical soundness is lacking in some parts of the paper.

    2.1. In Definition 2.1 you make reference to R_x but you do not define it. Other references are made to R_x in other places, but no formal definition is given.

    2.2. The conditional distribution presented in Definition 2.2 is not an actual distribution. If you simply zero-out some parts of the original distribution D_x using 1[r(z) = 1], then the D_x(z|r) will not integrate to 1. D_x(z|r) needs to be normalized by P[r(z) = 1] in order to be a proper probability measure. This also affects the definition of the precision (Definition 2.3). As D_x(z|r) is currently defined, the precision would evaluate to the probability of the output remaining the same (f(z) = f(x)) AND the rule firing (r(z) = 1), while the intended definition, I believe, is the probability of the output remaininig the same GIVEN that the rule fired.

    2.3. In Definition 2.5 you write "meeting the precision threshold τ with high probability δ", but the formal definition the follows indicates that the rule should meet the precision threshold τ with high probability *1 - δ*.

    2.4. The input domain in Section 4.1 for the task of income prediction, I believe, is wrongly defined. In my understanding, the input domain for the income prediction task is vectors with k features whose i-th feature takes values from F_i. In that case, the proper definition for X would be $X := \prod_{i=1}^{i \leq k} F_i$, that is the Cartesian product of the F_i's. The given definition uses the set union of the F_i's.

3. The experiments demonstrate that MAnchors produces rules with slightly lower precision and coverage and slightly larger length. I do not believe that this is detrimental, but the authors only hypothesize on what is causing this degradation. At the end of Section 4 the authors write: "In practice, users can mitigate the degradation of fidelity and understandability by configuring a higher similarity threshold τ_sim, albeit at the expense of prolonged cold-start latency". I think the value of τ_sim is a reasonable explanation for the degradation and increasing it would probably lead to better metrics, but this should also be backed up by an ablation study. I understand the space limitations and that the experiments they already provide are very computationally demanding (300 hours), but I insist that performing a small ablation study on the value value of τ_sim and including it in the Appendix would actually justify this claim. The authors could exclude Llama and YOLOv8 from the ablation study to make its demands reasonable, but still provide ample evidence for the effect of τ_sim on rule fidelity. I would rather they completely remove this claim, than present it with no evidence.


I also list some minor changes that in my opinion would improve the paper:

- Section 1, "and reduces 87% samples". The syntax here could be improved, perhaps "and reduced the samples needed by 87%".

- Beginning of Section 2.1. It would be better to reserve a bold font for vectors/tuples. You use a bold font for the individual features x_1, x_2, ... x_n, which is uncommon.

- Section 2.1, above definition 2.2. You write "Since our discussion does not involve the set of real numbers, all occurrences of $\mathbb{R}$ hereafter refer to the rule set". Consider switching from \mathbb to \mathcal to completely eliminate ambiguity. Use \mathcal for all domains, $\mathcal{X}, \mathcal{Y}, \mathcal{R}$.

- There are some other works that extend the Anchors algorithm that could be included in the Related Work section. Specifically:

    - "ReX: A Framework for Incorporating Temporal Information in Model-Agnostic Local Explanation Techniques" by Xin Zhang

    - "Temporal logic explanations for dynamic decision systems using anchors and Monte Carlo Tree Search" by Tzu-Yi Chiu et al.

    - "ConLUX: Concept-Based Local Unified Explanations" by Junhao Liu et al.

    - "Understanding Post-hoc Explainers: The Case of Anchors" by Gianluigi Lopardo et al.

---

> ### Author Rebuttal · Authors · 2026-03-29
>
> # Response to Reviewer zjXL
>
> We sincerely thank you for your careful and constructive review. Your insightful comments have helped us substantially improve both the clarity and rigor of our work. Below, we address each of your concerns in detail.
>
> ### 1. Reproducibility
> We fully agree that reproducibility is essential for impactful research. To promote transparency and facilitate verification, we have provided our implementation and experimental scripts in an anonymous repository:https://anonymous.4open.science/r/MAnchors. These materials are sufficient to reproduce all key results reported in the paper. In addition, we are committed to releasing a fully documented and publicly accessible version of the codebase upon acceptance, with the aim of supporting further research and practical adoption within the community.
>
> ### 2. Technical Formalism and Precision
> We appreciate your thorough examination of the technical presentation. In response, we have carefully revised the manuscript to improve precision and eliminate potential ambiguities:
>
> * **Notation:** We have replaced $\mathbb{R}$ with $\mathcal{R}$ throughout the manuscript to eliminate any potential ambiguity with the set of real numbers.
> * **Definitions:**
>     * A formal definition of $R_x$ is now provided prior to its first use.
>     * In **Definition 2.2**, we now normalize the conditional distribution by $P[r(z)=1]$. This ensures a proper probability measure and naturally resolves the precision definition in **2.3**.
>     * We have corrected the probability threshold in **Definition 2.5** to $1-\delta$.
> * **Input Domain:** In **Section 4.1**, we have redefined the input domain $X$ as a Cartesian product rather than a set union to accurately reflect its structure.
>
> ### 3. Impact of Similarity Threshold ($\tau_{sim}$)
> We agree that empirical evidence regarding the impact of $\tau_{sim}$ is essential to substantiate our findings. Following your suggestion, we conducted a targeted ablation study across multiple datasets.
>
> [Experimental Results](https://anonymous.4open.science/r/MAnchors/Sensitive.png)
>
> Due to time and computational constraints during the rebuttal period, this analysis focuses on our primary models and excludes Llama and YOLOv8. The results shows that increasing $\tau_{sim}$ effectively mitigates degradation in both understandability and fidelity. (The image only shows the experimental results for coverage; precision, due to the unchanged precision threshold setting, shows almost no difference and is therefore not shown here.) This provides empirical support for our claim that the similarity threshold plays a central role in governing the trade-off between these metrics.
>
> We will include these results, along with a more detailed discussion, in the Appendix of the revised manuscript.
>
> ### 4. Additional Revisions
>
> We also appreciate your suggestions regarding further improvements. In the revised version, we will:
> * Refine any imprecise or insufficiently formal language you mentioned; and
> * Incorporate and discuss the relevant works you highlighted that extend Anchors.
>
> ---
>
> We thank you again for your thoughtful and responsible review. Your input has been invaluable in improving the quality of this manuscript.

---

> > ### Author Rebuttal · Reviewer_zjXL · 2026-04-01
> >
> > My concerns have been addressed. I have raised my score to 5 (accept)

---

### Official Review · Reviewer_A6JC · 2026-03-11

**Soundness:** 2
**Presentation:** 2
**Significance:** 2
**Originality:** 3
**Overall Recommendation:** 3
**Confidence:** 4

**Summary:**

This paper proposes a method called MAnchors, which is a variant of Anchors but in a more efficient manner. Concretely, it saves a set of predicates from previous samples (in a shorter length) and computes explanations for new input making use of saved predicates from a similar sample. MAnchors uses Horizontal Transform and Vertical Transform to get the final explanation, which keeps the fidelity and understandability of the explanations.

**Compliance With Llm Reviewing Policy:**

Affirmed.

**Key Questions For Authors:**

(1) Why is the Sampling Reduction Ratio of LLaMA much higher than the other models? Isn't this metric related to the set of $τ_{p_{mid}}$ and $τ_{sim}$?

(2) Does the time consumption overhead of LLaMA come from inferencing the model itself, so that the speedup is significantly higher than the other models?

**Limitations:**

Please refer to the weaknesses.

**Strengths And Weaknesses:**

**Strengths**: The authors address the efficiency of explanation generation, which is a relevant research challenge in real-world applications. The experiment covers multiple data modalities such as tabular, textual, and image data. The evaluation also highlights the speedup effect of the proposed method.

**Weaknesses**:

(1) The algorithm design lacks practical use, and right now it is a naïve variant of Anchors. In practice, for a realistic application, a user wants to have real-time explanations. For this case, timely output matters. The authors could consider a pre-sampled dataset that tries to cover various inputs, so that at inference time, the focus of the algorithm is to provide real-time explanations compared to the current settings. It might be worth discussing the speedup of MAnchors compared to the original one under such a scenario.

(2) On the other hand, if explanations are not time-critical, the trade-off between memory and time needs to be further studied. The current setting is as follows: the speedup is about 2x for most of the cases, but the memory consumption is less discussed. For larger datasets, it is probably cheaper to parallelize the explanation generation to save time rather than using more memory.

(3) More ablation studies are needed to get more insights into the algorithm's strengths. For the set of $τ_{p_{mid}}$ and $τ_{sim}$, the performance of the algorithm (fidelity and understandability) vs. speedup should be added.

(4) Understandability of the explanations is also not thoroughly discussed. Currently, it is only related to the length.

(5) The presentation can be further improved. For instance, the name MAnchors is mentioned in the title but less so in the main paper. A workflow of MAnchors would be more helpful instead of the one for Anchors.

Overall, the current design of MAnchors is a bit raw. More experiments can show its strengths in various settings and address its practical usage.

---

> ### Author Rebuttal · Authors · 2026-03-29
>
> # Response to Reviewer A6JC
>
> We sincerely thank you for your thoughtful assessment and constructive feedback. Below, we address each of your concerns in detail and outline the corresponding improvements made to the manuscript.
> ## Weakness 1: Practical Utility and Pre-sampling
> We agree that real-time explanation capabilities are critical for practical deployment. To evaluate this, we investigated a pre-sampling approach where a representative dataset is processed offline. The results are summarized below:
> |Model|Task|Speedup (Pre-sampling Inputs=2000)|
> |--|--|--|
> |Llama|Text|10.21×|
> |Random Forest|Text|2.21×|
> |Random Forest|Tabular|2.71×|
> |NN|Tabular|1.98×|
> |GBT|Tabular|1.85×|
> |YOLOv8|Image|2.01×|
>
> While pre-sampling significantly accelerates the online phase, it introduces a prohibitive computational bottleneck offline. For instance, in our experiments with LLaMA models, this pre-sampling process required hundreds of GPU hours. MAnchors inherently addresses this limitation via a memoization-based strategy. This eliminates heavy upfront costs while continuously improving efficiency over time. Furthermore, MAnchors remains fully compatible with pre-training: setting $\tau_{sim}=0$ recovers a standard pre-sampling scheme.
> ## Weakness 2: Memory–Time Trade-off
> While MAnchors utilizes a "space-for-time" trade-off, empirical evidence demonstrates that the actual memory footprint is minimal.
>
> Excluding the memory occupied by the model and the dataset, the average memory overhead for MAnchors is **279.97MB**, compared to **276.81MB** for the original Anchors baseline. This marginal increase of approximately 3MB is a negligible cost for the substantial speedups achieved. To alleviate any reader concerns regarding resource consumption, we will include a dedicated discussion on space overhead in the revised manuscript.
> ## Weakness 3: Ablation Studies
> We agree that analyzing $\tau_{sim}$ and $\tau_{p_{mid}}$ is essential. Following your suggestion, we conducted ablation studies across multiple datasets:[Experimental Results](https://anonymous.4open.science/r/MAnchors/Sensitive.png)
>
> Due to the strict time constraints of the rebuttal period, this analysis focuses on primary models (excluding LLaMA and YOLOv8). The results demonstrate: $\tau_{sim}$ controls the trade-off between speedup and explanation quality (fidelity and interpretability). Higher values yield better fidelity and understandability but slightly reduce the overall speedup.$\tau_{p_{mid}}$ exhibits a smaller, inverse effect on these metrics. We will integrate these comprehensive results into the revision.
>
> ## Weakness 4: Metrics for Understandability
> We agree that evaluating understandability is important. However, to our knowledge, explanation length is the primary quantitative metric used in prior work (Qureshi, 2025). In our experiments, we observed that the lengths of explanation generated by MAnchors are largely unchanged compared to those produced by Anchors. The primary difference lies in the selection of predicates. To further investigate this, we conducted a manual comparison of the outputs from both methods. Our analysis suggests that these differences are not introduced by MAnchors, but rather caused by the inherent randomness of the Anchors—this variation also occurs when Anchors are run multiple times.
> Therefore, we conclude that MAnchors do not significantly reduce understandability. We will clarify and discuss this observation in the revised version.
>
> ## Weakness 5: Presentation Improvements
> We agree that the distinction between Anchors and MAnchors should be clearer. In the revision, we will:
> * Use the term “MAnchors” more consistently.
> * Add a [workflow of MAnchors](https://anonymous.4open.science/r/MAnchors/Workflow_of_MAnchors.png).
> ## Responses to Specific Questions
>
> **Q1: Why is the Sampling Reduction Ratio higher for LLaMA?**
>
> This primarily arises from the greater complexity of LLaMA’s decision process. Evidence from Anchors shows that LLaMA explanations require significantly more predicates (average length 4.4) compared to Random Forest on the same task (average length 1.6). Consequently, Anchors requires substantially more samples to construct explanations for LLaMA. In contrast, MAnchors can use memory and horizontal transformation to generate predicates via memory and horizontal transformation, thereby reducing the required number of samples more effectively for complex models like LLaMA.
>
> **Q2: What factors contribute to LLaMA’s overall speedup?**
>
> The significant acceleration in LLaMA is the result of two synergistic factors:
> 1. **High Baseline Latency:** Each inference step in LLaMA is computationally expensive. Reducing the number of samples via memoization yields a much higher absolute time saving compared to lighter models.
> 2. **Higher Complexity:** Due to the reasons explained in Q1.
> ---
> We believe these additions will address your concerns, and we are glad to provide further clarification if needed.

---

> > ### Author Rebuttal · Reviewer_A6JC · 2026-04-03
> >
> > I thank the authors for their detailed reply. However, I believe the paper still requires improvements, mainly in: (1) practical use cases and (2) overall presentation quality, to better serve the community.
> >
> > My main concerns remain focused on the practical implementation of this method. Memory costs should scale with the number of samples saved in memory. Consider a scenario involving millions of data points using a LLaMA model, the memory overhead saving these features for MAnchors should be significant. This is not directly comparable with Anchors, as Anchors' memory assumption is only a single data point. Therefore, I do not find W2 convincingly addressed.
> > Moreover, from a practical standpoint, the understandability concern W4 also remains. If Anchors is not the most human-friendly explanation method, nor the most real-time efficient one, then comparisons with alternative explanation methods become necessary to justify the proposed approach.
> >
> > New explanation methods are always welcome, but considering their impact in practice whether they produce usable explanations is becoming more important.

---

> > > ### Author Response · Authors · 2026-04-04
> > >
> > > Thank you for your thoughtful and constructive feedback. We appreciate your continued emphasis on the practical usability and clarity of our work. Below, we address your concerns in detail.
> > >
> > > ## (1)Practical Use Cases and Memory Overhead (W2)
> > > We thank the reviewer for raising this concern regarding the memory overhead in large-scale settings. We agree that if the memory footprint were to scale linearly with the number of data points, this would limit the practical applicability of the method. However, we would like to clarify that **this assumption does not hold for MAnchors**. The concern implicitly assumes that MAnchors stores explanations on a per-instance basis (i.e., one explanation per data point). In contrast, MAnchors is explicitly designed to avoid per-sample storage. Instead, it maintains a **shared set of high-coverage anchors**, which are reused across many inputs.
> > >
> > > Concretely, MAnchors stores only: (i) a KD-tree over sample embeddings, (ii) indices of samples in memory, and (iii) the corresponding feature subsets defining anchors. Importantly, these stored anchors act as reusable explanation units, rather than memorized per-instance results. As a result, the memory overhead should be understood as a form of compressed explanation cache, not a dataset-sized storage. This architecture fundamentally alters the scaling behavior.
> > >
> > > In practice, as demonstrated in the [memory size vs. dataset size plot](https://anonymous.4open.science/r/MAnchors/memory_size.png), we observe that the number of stored anchors grows very slowly and quickly saturates, as additional data points tend to reuse existing explanations instead of introducing new ones. This is analogous to rule-based models or clustering methods, where a relatively small number of representative structures can summarize a large dataset due to redundancy in local decision patterns. As a result, even at the scale of millions of inputs, the additional memory overhead introduced by MAnchors remains negligible compared to that of the model, the dataset, or even the storage requirements of the original Anchors method.
> > >
> > > Furthermore, the original Anchors method avoids memory overhead but recomputes explanations independently for each instance, leading to substantial redundancy in computation. MAnchors explicitly trades a small amount of memory for eliminating this redundancy.
> > >
> > > Therefore, the memory overhead of MAnchors should be viewed as a **bounded shared resource for reusable explanations**, rather than a cost that scales with the dataset size. We appreciate you highlighting this point, and we will revise the manuscript to clarify this distinction.
> > >
> > > ## (2)Understandability of Anchors (W4)
> > > We are pleased you recognize that MAnchors preserves the understandability of the Anchors. Regarding the premise that Anchors may not be the most human-friendly explanation method, we would like to gently push back and ground our design choice in recent literature. While no single method is universally optimal, **Anchors is widely regarded as one of the most human-friendly local explanation methods**, largely due to its intuitive, rule-based “if–then” format. Multiple recent empirical user studies strongly support this:
> > >
> > > >   * *"Transitioning to the human evaluation, the user study sheds light on complementary insights. Anchors emerge as the most useful explanation method... We conclude that Anchors produce the most useful explanations among the evaluated methods."* <br>— **Explainable AI for Text Classification: Lessons from a Comprehensive Evaluation of Post Hoc Methods (2024)**
> > > >   * *"Participants from the SSH group, in particular, proposed repositioning Anchor explanations from the end of the presentation to the middle. Their rationale was rooted in a preference for early clarity: they perceived anchors as both readable and cognitively accessible..."* <br>— **User-centric evaluation of explainability of AI with and for humans: A comprehensive empirical study (2025)**
> > > >   * *"Compared to attribution-based methods such as LIME and SHAP, Anchors offers more actionable (human-readable) rule-based explanations."* <br>— **Explainability in action: A metric-driven assessment of local explanations for healthcare tabular models (2025)**
> > >
> > > These findings affirm that Anchors holds a uniquely strong position regarding understandability. However, the primary reasons it has faced criticism—and the factors hindering its widespread adoption in the era of explaining large language models—are its relatively low efficiency and high sampling overhead.
> > >
> > > Importantly, the primary motivation behind MAnchors is not to claim superiority over all explanation methods, but rather to resolve the critical limitation of Anchors—its high computational cost and sampling inefficiency—while preserving its core strengths in accuracy and human readability.
> > >
> > > ---
> > > Thank you again for your feedback and suggestions. We hope the clarifications and evidence provided above fully address your concerns.

---

### Official Review · Reviewer_HT8P · 2026-03-11

**Soundness:** 4
**Presentation:** 3
**Significance:** 4
**Originality:** 3
**Overall Recommendation:** 5
**Confidence:** 4

**Summary:**

Anchors is good but quite slow. Most of the time is spent sampling neighboring data points and checking if the candidate explanations are good. Authors propose to cache in memory previously computed rules and samples to provide good initialization points when searching for candidate rules. The method described briefly consists in finding, if possible, a previously seen sample that is *similar* to the one desired to be explained, reusing an intermediate rule for that previous sample,  and adapting it to the current sample, and then running normal Anchors onwards.

Practical results show that, on three distinct target tasks, evaluated on six models, MAnchors roughly halves the processing time (up to eight times faster in some cases) and sampling ratio while retaining very high fidelity to the original Anchors rules. Depending on the task-model, MAnchors doesn't suffer at all from lower precision, lower coverage, or higher length.

**Compliance With Llm Reviewing Policy:**

Affirmed.

**Final Justification:**

As laid out in the Rebuttal Acknowledgement, the authors have cleared up some missing pieces and the submission is of better quality for it.

**Key Questions For Authors:**

> A vertical transformation, which incrementally refines this adapted rule to meet a required precision threshold.

Is that not the original Anchors method ? Later, authors (correctly) mention that it is, but the paper could make it clearer (and less stand out as a contribution) that VT is the previous method proposed by Anchors, using starting $r_1=\emptyset$

> Across all tasks, we set $\tau_p$ = 0.95 and $\delta$ = 0.6 to remain consistent with the default parameters of the Anchors method. The additional parameters introduced by our approach are set to $\tau_{p_{mid}}$  = 0.8 and $\tau_{sim}$  = 0.6

While it is evident that using the same $\tau_p$ and $\delta$ is absolutely necessary to fairly compare Anchors and MAnchors, is it "fair" to use the same hyperparameters for distinctly different dataset/target task/modality ?

Q: what about Beam search (method used in original Anchors) ? How is $r_{mid}$ determined then ? Could your method support it ?

Q: The function Dist(.) seems like a huge bottleneck of the proposed method, how do authors remedy the eventual problem entailed by a "mediocre" Dist(.) function ? It is a bit unclear _how_ Dist(.) identifies (dis)similar features.

Q: The proposed method adds yet another **two** hyperparameter to tune for, how do you propose to find/choose a good value for $\tau_{p_{mid}}$ ?

Q: Likewise, how do you propose to find a good $\tau_{sim}$ ?

**Limitations:**

yes

**Strengths And Weaknesses:**

# Strengths

* The submission is rather well written, easy to understand, and properly reintroduces Anchors. Authors clearly show where they deviate from the original implementation, and how they improve the existing algorithm.
* The idea itself is quite simple to grasp, and fills a gap in the non-formal XAI literature.
* A theoretical complexity analysis is provided
* The practical results are done on a range of tasks and models (Llama 3-8B, YOLOv8, ...) and results support the claims made by the authors.

# Weakness

* Authors propose a method to reduce processing time by adding cache-memory and provide a time complexity analysis **but** do not do a memory complexity analysis. An opportunity missed in my opinion. Furthermore, the time complexity analysis does not take into account  the complexity aspect of HT, e.g. calling Dist(.) repeatedly $j$ times where $j$ is the (expected) size of the cached Anchors rule hit.
* Too little mention of KL-LUCB even though a key metric measures is the sampling reduction ratio.
* No mention of beam search. It is a strong contribution from the original Anchors paper, and the authors don't :
   - compare against it
   - comment on how beam search is supported (or not) by MAnchors
* No mention of how to find/choose important hyperparameters. I would expect at least one hyperparameter sweep / bayesian optimisation to make out a "rule of thumb" on how to choose the correct $\tau_{p_{mid}}$ or $\tau_{sim}$
* A few typos, the presentation could be polished. Authors should make sure the pseudocodes and the mathematical definitions are "clean".


#### Typos
- local,even ← missing space
- Def 2.1 overflows
- Def 2.5/3.2 "Coverage" in math font + should be simply max ?
- "(Line 10)" overflows
- "time-consuming, We"
- Algorithm 1 and SpeedUp formula says MAnchors takes  $\mathbf{x}$ and $f$ as inputs, $\tau_p$ , $\tau_{p_{mid}}$ and $\tau_{sim}$, but $\mathbb M$ is also an important (tracked) parameter ! → The method will have wildly different results depending on what is contained in $\mathbb M$ at the moment $\mathbf{x}$ is processed.

---

> ### Author Rebuttal · Authors · 2026-03-30
>
> # Response to reviewer HT8P
> We sincerely thank you for your careful and constructive review. Your insightful comments have helped us substantially improve both the clarity and rigor of our work. Below, we address each of your concerns in detail.
>
> ## Space Complexity Analysis
> We agree that analyzing space complexity is necessary, especially since our method trades space for time. The additional memory mainly comes from: a) the KD-tree for embeddings, b) stored sample indices, and c) associated Anchor indices. Let $n$ be the number of samples, $m$ the average number of features, and $k$ the embedding dimension. The overall overhead is $O(nk) + O(n) + O(nm)$, which is negligible relative to the dataset size. Empirically, MAnchors requires **279.97MB** on average, compared to **276.81MB** for Anchors (excluding model and dataset memory). This marginal increase is a reasonable trade-off for the observed speedup. We will include this analysis in the revised manuscript.
>
> ## Time Complexity Analysis
> We did not analyze HT separately because both HT and VT are triggered under memory hits, where the dominant cost arises from VT. Since VT involves iterative perturbations whose iterations depend on model behavior, its time complexity is difficult to express. Thus, analyzing the lighter HT stage alone offers limited insight. Although HT requires multiple calls to the Dist function, its cost remains bounded and negligible compared to VT. If $n_1$ is the number of predicates in Anchors for the memory input and $n_2$ the number of features in the online input, HT requires at most $n_1 × n_2$ Dist calls. Notably, Dist shares the same model as the perturbation process in VT (and Anchors), where the number of calls is much larger than $n_1 × n_2$. Therefore, HT introduces only minor overhead. We will add a brief discussion to clarify this point.
>
>
> ## Relationship Between VT and Beam Search
> While VT shares structural similarities with the original Beam Search used in Anchors, they are not strictly identical. Technically, VT does not start from an empty predicate set. Instead, it includes an optional pre-filtering step that removes low-contribution predicates via light sampling. This detail was omitted in the original manuscript as an implementation nuance, but we will clarify it to avoid confusion. More importantly, this design reflects a non-intrusive modification philosophy: by preserving the overall search framework of Beam Search, VT maintains the interpretability quality of Anchors while improving efficiency. In addition, although both methods employ the KL-LUCB algorithm for precision estimation, VT reduces the number of search iterations, which in turn decreases the total sampling required by KL-LUCB. We will further elaborate on this interaction in the revision to make the distinction clearer.
>
>
> ## Configuration of Parameters $\tau_p$ and $\delta$
> These parameters are inherited from Anchors. Since Anchors does not define dataset-specific settings, we adopt its default values for fairness. In practice, their values depend on user requirements, particularly the desired explanation precision. Thus, defining a single “fair” configuration across datasets is difficult.
>
> ## Selection of Optimal $\tau_{sim}$ and $\tau_{p_{mid}}$ Parameters
> We appreciate this valuable suggestion. Following your recommendation, we conducted a hyperparameter analysis of $\tau_{sim}$ and $\tau_{p_{mid}}$. The [results](https://anonymous.4open.science/r/MAnchors/Sensitive.png) reveal a clear trade-off between efficiency and explanation quality. Increasing $\tau_{sim}$ generally improves coverage, as stricter similarity constraints lead to more precise rule selection, but at the cost of longer runtime. Conversely, increasing $\tau_{p_{mid}}$ reduces runtime by saving more predicates, with a slight reduction in coverage.
> Notably, we observe a stable region where high coverage is maintained while runtime and explanation length remain moderate. Based on these findings, we recommend:$$\tau_{sim} \in [0.6, 0.8], \quad \tau_{p_{mid}} \in [0.7, 0.8],$$which provides a balanced trade-off between efficiency and explanation quality.
>
> ## `Dist` function
>
> The `Dist` function employs the same model utilized internally by Anchors during feature perturbation—specifically, when Anchors perturbs a feature, it uses a model to identify similar words for that token. Consequently, our method adopts this consistent model for embedding purposes and measures feature similarity based on the distances between these embeddings. To mitigate potential inaccuracies, we apply an additional filtering step after HT, removing predicates with low contribution through sampling. This serves as a safeguard when similarity estimates are imperfect.
>
> ---
>
> We are grateful for the reviewer's time and expertise in evaluating our submission. We believe that addressing these points will significantly strengthen the paper and we look forward to incorporating these changes in the final version.

---

> > ### Author Rebuttal · Reviewer_HT8P · 2026-04-03
> >
> > The authors have taken the time to answer my questions. I appreciate that they will mention more the memory and time complexities in the final version, as laid out in their rebuttal. I am still unsure whether Beam Search is possible with VT, and if it is possible, perhaps just a small word on how to do so.
> >
> > I have changed my score to (5) accept.

---

### Official Review · Reviewer_v3b2 · 2026-03-13

**Soundness:** 3
**Presentation:** 3
**Significance:** 3
**Originality:** 2
**Overall Recommendation:** 5
**Confidence:** 3

**Summary:**

This paper proposes a method, "MAnchors ", to improve the efficiency of "Anchors", a post-hoc model-agnostic explanation method. The authors propose retaining intermediate rules from previously examined input instances and using them to reduce explanation generation time for subsequent input instances. When a new input comes, MAnchors checks if explanations have been generated for similar input instances before and starts rule generation from the intermediate rules of that instance instead of starting from scratch. Experiments have been conducted for tabular, text and image data using various ML models, including LLMs. The experimental results demonstrate that the proposed method can speed up rule generation by at least 1.5 times in each setting, and notably, when using LLMs for sentiment analysis, it can improve speed by more than 8 times.

**Compliance With Llm Reviewing Policy:**

Affirmed.

**Final Justification:**

The authors have provided convincing results addressing my concerns. Thus, I have raised the score to 5 (accept).

**Key Questions For Authors:**

1. As per the Fidelity results provided in the Appendix, the MAnchors fall short in coverage compared to the Anchors. Can you explain the reason for this? Also, the authors are encouraged to include fidelity results for methods and datasets in the main body, as these results are important for evaluating the methods.
2. Can you provide results for more LLM usage to strengthen the argument that Anchors significantly reduce explanation generation for LLMs? You may present results when different LLMs are used for sentiment analysis, other NLP tasks and image classification.
3. Can you also provide results for different image classification architectures, particularly transformer-based approaches?

**Limitations:**

Authors are encouraged to include a section discussing limitations and future work.

**Strengths And Weaknesses:**

Soundness: The proposed method is technically correct. The claim of reducing the computational time of explanation generation while maintaining explanation quality has been supported through experiments conducted on tabular, text, and image data using various ML models. However, the experiments can be expanded to demonstrate the generalizability of the results. In particular, the proposed method can speed up anchor generation for LLMs by 8 times. However, the experiments have been conducted only using the Llama model applied for sentiment analysis. More experiments should be presented to demonstrate whether the speedup by MAnchors is general to LLMs and whether the same improvement can be obtained when LLMs are applied to other data types. Similarly, authors are encouraged to provide more experiments for image classification using various classification models.

Presentation: The paper is generally well written. I enjoyed reading the paper. The authors clearly state the limitations of the existing Anchors method and explain the proposed method in detail.

Significance: The paper addresses an important research question. The time required to generate explanations is also crucial when deploying XAI methods in the real world. According to the experimental results, the proposed MAnchors method has significantly increased the speed of explanation generation. Hence, MAnchors has the potential to increase the use of the Anchors method for explanation generation, as it significantly reduces the time required.

Originality: The paper proposes a new method to reduce the computational time of the existing XAI method, Anchors, and achieves significant improvements over it. The authors have explained the design decisions behind the proposed method.

---

> ### Author Rebuttal · Authors · 2026-03-29
>
> # Response to reviewer v3b2
> Thank you for your thoughtful and constructive feedback. We are greatly encouraged by your positive assessment of the technical soundness and practical significance of our work. Below, we address your specific questions and detail the new experiments conducted during the rebuttal period.
>
> 1. **Coverage Degradation and Fidelity (Addressing Q1)**
> We acknowledge that the explanation coverage of MAnchors is slightly lower than that of Anchors. This behavior stems from the Horizontal Transformation (HT) process: when similar inputs retrieved from memory are not sufficiently similar to the current input, some generated predicates may contribute less precision. In such case, MAnchors introduce additional predicates to maintain precision threshold, which increases predicates of output and can lead to a modest reduction in coverage. To solve this, MAnchors provides two hyperparameters, $\tau_{\text{sim}}$ and $\tau_{\text{pmid}}$, that allow users to balance between speedup and coverage. In our experiments, we select these parameters to maximize speedup while constraining the coverage reduction to within 10%. Detailed settings are provided in Section 4.1.
> Regarding the presentation of whole fidelity data in the paper, we strongly agree with your point. Due to space limitations, we currently only present the Fidelity of Llama, which shows the most significant speedup, in the main text; the others are placed in the appendix. If this paper is accepted, we will include complete Fidelity data for all models and datasets in the main text.
>
> 2. **Additional LLM Experiments (Addressing Q2)**
> We agree that broader evaluation is necessary to demonstrate the generalizability of our method. Our initial submission focused heavily on sentiment analysis due to the prohibitive computational cost of generating the Anchors baseline (approximately 300 hours per experiment). To strengthen our claims, we have now conducted additional experiments using the Gemma-2-2B model across multiple text classification tasks, specifically SST-2 (sentiment analysis) and AG News (news classification).
>
> |Dataset|Anchors Avg.Time(s)|MAnchors Avg.Time(s)|Speedup|Sampling Reduction Ratio| Anchors Avg.Coverage|MAnchors Avg.Coverage|Anchors Avg.Length|MAnchors Avg.Length|
> |--|--|--|--|--|--|--|--|--|
> |SST-2|269.96(±55.07)|59.17(±22.87)|4.56×|78%|0.17(±0.15)|0.13(±0.10)|4.6(±2.1)|5.2(±2.3)|
> |Ag_news|259.31(±135.28)|88.96(±49.07)|2.91×|66%|0.50(±0.02)|0.48(±0.02)|1.0(±0.1)|1.1(±0.1)|
>
> These new results demonstrate that our approach consistently achieves substantial acceleration across different datasets while preserving explanation fidelity. While the relative speedup ratios are naturally bounded by the increased complexity of the Gemma-2-2B architecture compared to our earlier small-model experiments, the massive reduction in sampling strongly validates the method's efficiency in complex LLM settings.
>
> 3. **Transformer Architectures and Image Classification (Addressing Q3)**
> We appreciate the reviewer highlighting the importance of testing diverse architectures, particularly transformer-based models. Generating the exact Anchors baseline for large-scale Vision Transformers (ViTs) is computationally bottlenecked within the short rebuttal window. However, to directly address the core of your concern regarding transformer compatibility, we evaluated our method on a fine-tuned Gemma-2-2B.
>
> |Anchors Avg.Time(s)|MAnchors Avg.Time(s)|Speedup|Sampling Reduction Ratio|Anchors Avg.Coverage|MAnchors Avg.Coverage|Anchors Avg.Length|MAnchors Avg.Length|
> |--|--|--|--|--|--|--|--|
> |105.87(±27.71)|45.97(±11.54)|2.30×|58%|0.05(±0.02)|0.04(±0.02)|2.4(±0.2)|2.7(±0.4)|
>
> The consistent acceleration and sampling reduction observed on this LLM architecture provide strong empirical evidence that our method effectively generalizes to transformer self-attention and MLP blocks. This gives us high confidence that our approach will similarly generalize to ViTs, which we have prioritized as our immediate next step for future work.
>
> 4. **Limitations and Future Work**
> Following your valuable suggestion, we will add a dedicated "Limitations and Future Work" section: While users can balance speedup and fidelity via $\tau_{\text{pmid}}$ and $\tau_{\text{sim}}$, the slight decline in fidelity remains a limitation in high-stakes scenarios requiring exactness. To address this, future work will explore advanced alignment strategies that leverage memory samples even when they lack high embedding-space proximity to the current input. Specifically, we will investigate learning a parameterized mapping between memory inputs and online inputs to capture underlying functional equivalences beyond simple distance metrics.
>
> We thank the reviewer again for the insightful comments. We believe the additional experiments and clarifications significantly strengthen the paper and thoroughly address the concerns regarding generalizability, fidelity, and completeness.

---

> > ### Author Rebuttal · Reviewer_v3b2 · 2026-04-02
> >
> > The authors have provided convincing results addressing my concerns. Thus, I have raised the score to 5 (accept).

---

### Decision · Program_Chairs · 2026-04-30

**Decision:**

Accept (regular)

**Comment:**

This paper on local, rule-based model-agnostic explanations received four reviews, resulting in an average recommendation score of 4.5. The rebuttal phase was very productive, allowing the authors to clarify several points and ultimately improve the paper's average score. At this stage, I recommend acceptance. However, I strongly encourage the authors to address the reviewers' comments, particularly regarding technical quality, parameter analysis, memory overhead, and the clarity of the explanations, in a revised version of the paper.